# Learning Transferable Features for Implicit Neural Representations

**Kushal Vyas**
kushal.vyas@rice.edu

**Ahmed Imtiaz Humayun**
imtiaz@rice.edu

**Aniket Dashpute**
aniket.dashpute@rice.edu

**Richard G. Baraniuk**
richb@rice.edu

**Ashok Veeraraghavan**
vashok@rice.edu

**Guha Balakrishnan**
guha@rice.edu

Rice University

## Abstract

Implicit neural representations (INRs) have demonstrated success in a variety of applications, including inverse problems and neural rendering. An INR is typically trained to capture one signal of interest, resulting in learned neural features that are highly attuned to that signal. Assumed to be less *generalizable*, we explore the aspect of transferability of such learned neural features for fitting similar signals. We introduce a new INR training framework, STRAINER that learns transferrable features for fitting INRs to new signals from a given distribution, faster and with better reconstruction quality. Owing to the sequential layer-wise affine operations in an INR, we propose to learn transferable representations by sharing initial *encoder* layers across multiple INRs with independent *decoder* layers. At test time, the learned *encoder* representations are transferred as initialization for an otherwise randomly initialized INR. We find STRAINER to yield extremely powerful initialization for fitting images from the same domain and allow for a $\approx +10dB$ gain in signal quality early on compared to an untrained INR itself. STRAINER also provides a simple way to encode data-driven priors in INRs. We evaluate STRAINER on multiple in-domain and out-of-domain signal fitting tasks and inverse problems and further provide detailed analysis and discussion on the transferability of STRAINER's features. Our demo can be accessed here.

## 1 Introduction

Implicit neural representations (INRs) are a powerful family of continuous learned function approximators for signal data that are implemented using multilayer perceptron (MLP) deep neural networks. An INR $f_\theta : \mathbb{R}^m \mapsto \mathbb{R}^n$ maps *coordinates* lying in a $m$-dimensional space to a value in a $n$-dimensional output space, where $\theta$ represents the MLP's tunable parameters. For example, a typical INR for a natural image would use an input space in $\mathbb{R}^2$ (consisting of the $x$ and $y$ pixel coordinates), and an output space in $\mathbb{R}^3$ (representing the RGB value of a pixel). INRs have demonstrated several useful properties including capturing details at all spatial frequencies [39, 36], providing powerful priors for natural signals [36, 39], and facilitating compression [12, 27]. For these reasons, in the past 5 years, INRs have found important uses in image and signal processing including shape representation [17, 16], novel view synthesis [31, 34, 42], material rendering [24], computational imaging [5, 30], medical imaging [49], linear inverse problems [8, 44], virtual reality [11] and compression [12, 27, 43, 51].

38th Conference on Neural Information Processing Systems (NeurIPS 2024).

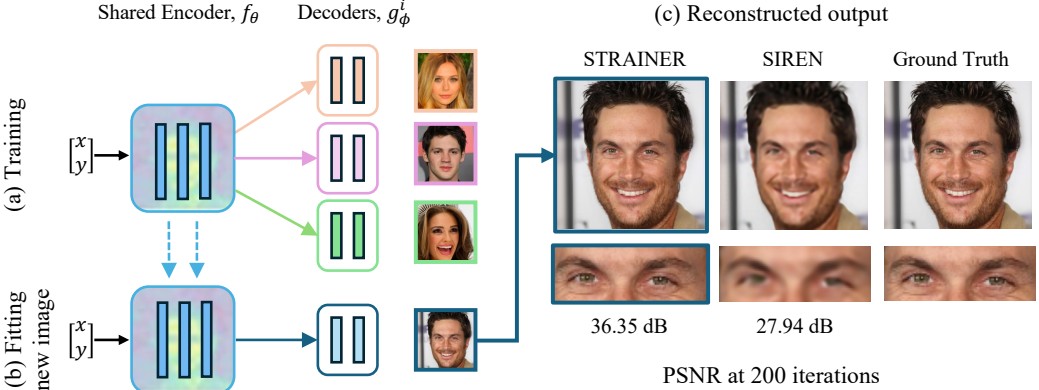

Figure 1: **STRAINER** - **Learning Transferable Features for Implicit Neural Representations.**
During training time (a), STRAINER divides an INR into *encoder* and *decoder layers*. STRAINER fits
similar signals while sharing the encoder layers, capturing a rich set of transferrable features. At test-
time, STRAINER serves as powerful initialization for fitting a new signal (b). An INR initialized with
STRAINER's learned encoder features achieves (c) faster convergence and better quality reconstruction
compared to baseline SIREN models.

A key difference between training INRs compared to other neural architectures like CNNs and
Transformers is that a single INR is trained on a single signal. The features learned by an INR,
therefore, are finely tuned to the morphology of just the one signal it represents. SplineCAM [19]
shows that INRs learn to finely partition the input coordinate space by essentially overfitting to
the spatial gradients (edges) of the signal. While this allows an INR to represent its signal with
high fidelity, its features can not "transfer" in any way to represent a second signal, even with
similar content. If INRs could exhibit elements of transfer learning, as is the case with CNNs and
Transformers, their potential would dramatically increase, such as by encoding data distribution
priors for inverse imaging problems.

In this work, we take a closer look at INRs and transferable features, and demonstrate that the *first
several layers* of an INR can be readily transferred from one signal to another from a domain when
trained in a shared setting. We propose STRAINER , a simple INR training framework to do so (see
Figure 1). STRAINER separates an INR into two parts: an "encoder" that maps coordinates to features,
and a "decoder" that maps those features to output values. We fit the encoder over a number of
training signals (1 to 10 in our experiments) from a domain, e.g., face images, with separate decoders
for each signal. At test time, we initialize a new INR for the test signal consisting of the trained
encoder and a randomly initialized decoder. This INR may then be further optimized according to the
application of interest. STRAINER offers a simple and general means of encoding data-driven priors
into an INR's parameter initialization.

We empirically evaluate STRAINER in several ways. First, we test STRAINER on image fitting
across several datasets including faces (CelebA-HQ) and medical images (OASIS-MRI) and show
(Figure 2) that STRAINER's learned features are indeed transferrable resulting in a $\approx$+10dB gain in
reconstruction quality compared to a vanilla SIREN model . We further assess the data-driven prior
captured by STRAINER by evaluating it on inverse problems such as denoising and super resolution.
Lastly, we provide a detailed exploration of how STRAINER learns transferable features by exploring
INR training dynamics. We conclude by discussing consequences of our results for the new area of
INR feature transfer.

## 2 Background

**Implicit neural representations.** We define $f_\theta(p)$ as an implicit neural representation (INR)
[31, 39, 29] where $f_\theta$ is a multi-layer perceptron (MLP) with randomly initialized weights $\theta$ and $p$ is
the $m$-dimensional coordinates for the signal. Each layer in the MLP is an affine operation followed

by a nonlinearity such as ReLU [31], or sine [39]. Given an $n$-dimensional signal $I(p)$, the INR learns a mapping $f : \mathcal{R}^m \to \mathcal{R}^n$. The INR is iteratively trained to fit the signal by minimizing a loss function such as $L_2$ loss between the signal $I(p)$ and its estimate $f_\theta(p)$:

$$\theta^* = \arg\min_\theta \sum_i \| \mathcal{A}f_\theta(p_i) - I(p_i) \|_2^2 \ , \tag{1}$$

where $p_i$'s $\in \mathcal{R}^m$ span the given coordinates, $\theta^*$ are the optimal weights that represent the signal, and $\mathcal{A}$ is a differentiable forward model operator such as identity for signal representation and a downsampling operation for inverse problems such as super-resolution.

**Representation capacity of INRs.** The representation capacity of an INR can be described as the quality of signal the INR can represent within some number of iterations. ReLU-based INRs suffer from spectral bias during training[26], preferentially learning low frequency details in a signal and thus leading to a blurry reconstruction of the represented signal. Fourier positional encodings [21, 31, 46] or sinusoidal activation functions (SIREN) [39] help better capture high frequency information.

Recent works increase the representation capacity of INRs with activations flexible in the frequency domain. WIRE [36] uses a continuous Gabor wavelet-based nonlinearity, and demonstrates impressive results for a range of forward and inverse problems. FINER [26] splits the sine nonlinearity to have a flexible frequency coverage, and DINER[50] uses a hash map to non-uniformly map the input coordinates to a feature vector, effectively re-arranging the spatial arrangement of frequencies and leading to faster and better reconstruction quality.

**Weight initialization for INRs.** Previous work has shown that smart initialization of INR weights allows for faster convergence. As shown in the SIREN study [39], hyper-networks are pro-

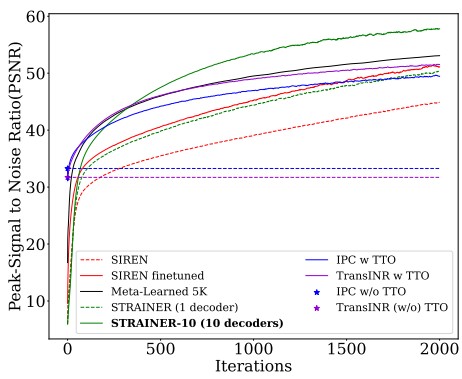

Figure 2: **STRAINER learns faster.** We show the reconstruction quality (PSNR) of different initialization schemes for in-domain image fitting on CelebA-HQ [22]. We compare SIREN [39] model initialized by (1) random weights (SIREN), (2) fitting on another face image (SIREN finetuned), (3) STRAINER -1 (trained using one face image), and (4) STRAINER-10 (trained using ten face images). We also evaluate against multiple baselines such as Meta-Learned 5K [45], TransINR[9], and IPC[23]

posed to capture a prior over the space of implicit functions, mapping a random code to the weights of a SIREN model. Further, TransINR[9] also shows Transformer-hypernetworks as powerful metalearners for INR weight initialization. MetaSDF [38] and Light Field Networks (LFN) [40] use meta-learning-based initialization schemes to fit signed distance functions (SDFs) and light fields. Neural Compression algorithms [14, 43] use weights obtained from meta-learning optimization as a reference to store MLP weights, leading to better compression than naively storing MLP weights. Tancik et al. [45] propose meta-learning-based approaches such as Model-Agnostic Meta-Learning (MAML)[15] and Reptile[32] for coordinate based neural representations. While these meta-learning approaches yield powerful initialization, they often require long computation time (over 150K steps [45]) and ample numbers of training data, and are unstable to train [48]. Further, meta-learning initial modulations for an INR which are later optimized to fit data within few gradient updates has been shown to be an effective and scalable[6] strategy for smoothly representing data(sets) as functa(sets)[13]. Contrary to our approach, Implicit Pattern Composers (IPC)[23] proposes to keep the second layer of an INR instance-specific, while sharing the remaining layers and use a transformer hypernetwork to learn the modulations for the INR.

**Prior informed INRs.** Recent work has also explored embedding a learned prior in INRs for tasks such as audio compression, noise removal, and better CT reconstructions. Siamese Siren [25] uses a similar approach where they propose a compact siamese INR whose initial layers are shared followed by 2 siamese decoders. Since 2 randomly initialized decoders will yield slightly different reconstructions, this difference is leveraged for better noise estimation in audio signal. NERP [37] learns an internal INR prior for medical imaging by first fitting high quality MRI or CT data. Weights

of this learned INR are used as an initialization for reconstructing new MRI or CT undersampled data. While this paper shows a method to learn an implicit prior, their prior embedding is learned from a single MRI or CT scan of the same subject whereas our work explores learning a prior for INRs by constraining it to learn an underlying implicit representation across multiple different images. PINER [41] introduces a test-time INR adaptation framework for sparse-view CT reconstruction with unknown noise.

## 3 Methods

We introduce STRAINER . We first explain our motivation to share initial layers in an INR Section 3.1. In Section 3.2 we describe the training phase of STRAINER where we learn transferrable features for INRs by sharing the initial layers of $N$ INRs being fit independently to $N$ images. Section 3.3, details how our captured basis is used to fit an unseen image. In subsequent sections, we seek to understand what our shared basis captures and how to expand it to other problems such as super resolution. For simplicity, we build upon the SIREN [39] model as our base model.

### 3.1 Why share the initial INR layers?

A recent method called SplineCAM [19] provides a lens with which to visualize neural network partition geometries. SplineCAM interprets an INR as a function that progressively warps the input space and fits a given signal through layerwise affine transforms and non-linearities [19]. For continuous piecewise affine activation functions, we use an approximation to visualize (see Figure 6) the deep network's partition geometry for different pre-activation level sets [20].

An INR fit to a signal highly adapts to the underlying structure of the data in a layer-wise fashion. Furthermore, by approximating the spatial position of the pre-activation zero level sets, we see that initial layers showcase a coarse, less-partitioned structure while deeper layers induce dense partitioning collocated with sharp changes in the image. Since natural signals tend to be similar in their lower frequencies, we hypothesize that initial layers of multiple INRs are better suited for *transferability*. We therefore design STRAINER to share the initial *encoder* layers, effectively giving rise to an input space partition that can generalize well across different similar signals.

### 3.2 Learning transferable features from $N$ images

Consider a SIREN [39] model $h(p)$ with $L$ layers. Let $K$ out of $L$ layers correspond to an *encoder* sub-network represented as $f_\theta$ The remaining layers correspond to the *decoder* sub-network represented as $g_\phi$ as seen in Figure 1(a). For given input coordinates $p$, we express the SIREN model $h_{\phi,\theta}(p)$ as a composition ($\circ$) of our encoder-decoder sub-networks.

$$h_{\phi,\theta}(p) = g_\phi \circ f_\theta(p) \,, \tag{2}$$

In a typical case, given the task of fitting $N$ signals, each of the $N$ signals is independently fit to an INR, thus not leveraging any similarity across these images. Since we want to learn a shared representation transferrable across multiple similar images, our method shares the encoder $f_\theta$ across all $N$ INRs while maintaining a set of individual signal-specific decoders $g_\phi^1 \ldots g_\phi^N$. Our overall architecture is shown in Figure 1. We call this STRAINER's training phase - Figure 1(a). We start with randomly initialized layers and optimize the weights to fit $N$ signals in parallel. For each signal $I_i(p)$, we use a $L_2$ loss between $I_i(p)$ and its corresponding learned estimate $h_{\phi,\theta}^i(p)$ and sum the loss over all the $N$ signals. Iteratively, we learn a set of weights $\Theta$ that minimizes the following objective:

$$\Theta^* = \arg\min_\Theta \sum_{i=1}^{N} || \, g_\phi^i \circ f_\theta(p) - I_i(p) \, ||_2^2 \,, \tag{3}$$

where $\Theta = [\theta, \phi^1 \ldots \phi^N]$ represents the full set of weights of the shared encoder ($\theta$) and the $N$ different decoders ($g_\phi^1 \ldots g_\phi^N$) and $\Theta^*$ represents the resulting optimized weights.

### 3.3 Fitting an unseen signal with STRAINER

After sufficient iterations during STRAINER's training phase, we get optimized encoder weights $f_{\theta^*}$ which corresponds to the rich shared representation learned over signals of the same category. To fit a

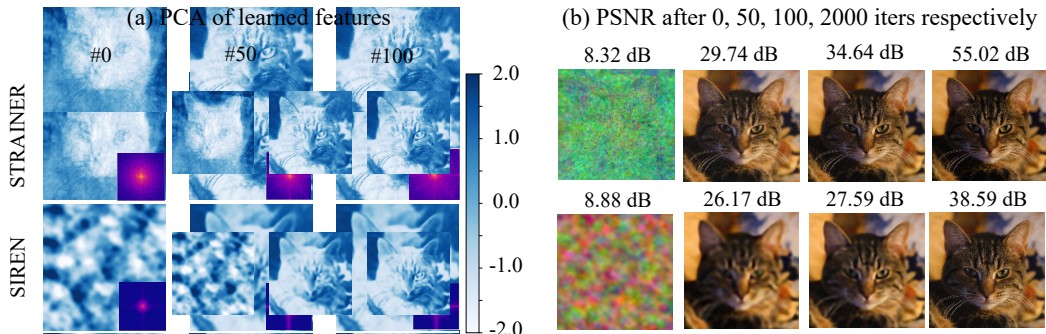

Figure 3: **Visualization of learned features in STRAINER and baseline SIREN model**. We visualize (a) the first principal component of the learned encoder features for STRAINER and corresponding layer for SIREN . At iteration 0, STRAINER's features already capture a low dimensional structure allowing it to quickly adapt to the cat image. High frequency detail emerges in STRAINER's learned features by iteration 50, whereas SIREN is lacking at iteration 100. The inset showing the power spectrum of the reconstructed image further confirms that STRAINER learns high frequency faster. We also show the (b) reconstructed images and remark that STRAINER fits high frequencies faster.

novel signal $I_\psi(p)$ we initialize the STRAINER model with the learned shared encoder weights $f_{\theta=\theta^*}$ and randomly initialize decoder $g_\phi^\psi$ weights to solve for:

$$\phi^*, \theta^* = \arg\min_{\phi,\theta} \| g_\phi^\psi \circ f_{\theta=\theta^*}(p) - I_\psi(p) \|_2^2 . \tag{4}$$

$f_{\theta=\theta^*}$ serves as a learned initial encoder features. Our formulation is equivalent to a initial set of learned encoder features followed by a set of random projections. While fitting an unseen signal, we iteratively update all the weights of the STRAINER model, similar to any INR.

### 3.4 Learning an intermediate partition space in the shared encoder $f_{\theta^*}$

During the training phase, explicitly sharing layers in STRAINER allows us to learn a set of INR features which exhibits a common partition space shared across multiple images. Since deep networks perform layer wise subdivision of the input space, sharing the encoder enforces the layer to find the partition that can be further subdivided into multiple coarse partitions corresponding to the tasks being trained. In Figure 6(a.ii), while pre-training an INR using the STRAINER framework on CelebA-HQ dataset, we see emergence of a face-like structure captured in our STRAINER encoder $f_{\theta^*}$. We expect our STRAINER encoder weights $f_{\theta^*}$ to be used as transferrable features and be used as initialization for fitting unseen in-domain samples.

In comparison, meta learning methods to learn initialization for INRs[45] exhibit a partitioning of the input space that is closer to random. As seen in Figure 6(a.i) there is a faint image structure captured by the the learned initialization. This is an indication that the initial subdivision of the input space, found by the meta learned pre-training layers, captures less of the domain specific information therefore is a worse initialization compared to STRAINER . We further explain our findings in Section 5 and discuss STRAINER's learned features being more transferrable and lead to better quality reconstruction.

## 4   Experiments

In all experiments, we used the SIREN [39] MLP with 6 layers and sinusoid nonlinearities. We considered two versions of STRAINER : (i) STRAINER (1 decoder), where the encoder layers are initialized using our shared encoder trained on a single image, and (ii) STRAINER-10 (10 Decoders), where the encoder layers are initialized using our shared encoder trained on 10 images. We considered the following baselines: (i) a vanilla SIREN model with random uniform initialization [39], (ii) a fine-tuned SIREN model initialized using the weights from another SIREN fit to an image from the same domain, (iii) a SIREN model initialized using Meta-learned 5K [45], (iv) transformer-based metalearning models such as TransINR[9] and IPC[23]. We ensured an equal number of learnable

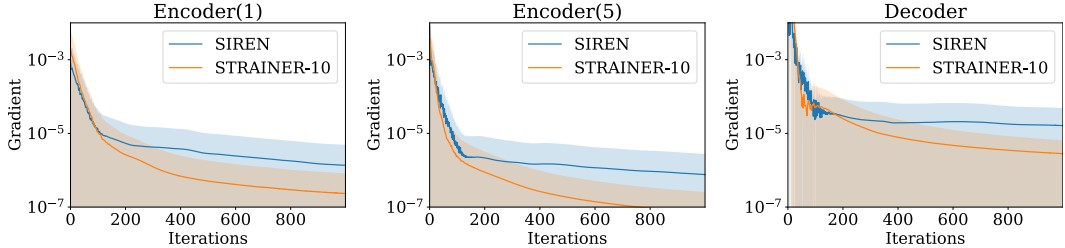

Figure 4: **STRAINER converges to low and high frequencies fast.** We plot the histogram of absolute gradients of layers 1,5 and last over 1000 iterations while fitting an unseen signal. At test time, STRAINER's initialization quickly learns low frequency, receiving large gradients update at the start in its initial layers and reaching convergence. The Decoder layer in STRAINER also fits high frequency faster. Large gradients from corresponding SIREN layers show it learning significant features as late as 1000 iterations.

parameters (neurons) for all models. We normalized all images between (0-1), and input coordinates between (-1,1). We used the Adam optimizer with a learning rate of $10^{-4}$ for STRAINER's training and test-time evaluation, unless mentioned otherwise. Further implementation details are provided in Supplementary.

## 4.1 Datasets

We mainly used the CelebA-HQ [22], Animal Faces-HQ (AFHQ) [10], and OASIS-MRI [18, 28] images for our experiments. We randomly divided CelebA-HQ into 10 train images and 550 test images. For AFHQ, we used only the cat data, and used ten images for training and 368 images for testing. For OASIS-MRI, we used 10 of the (template-aligned) 2D raw MRI slices for training, and 144 for testing. We also used Stanford Cars[2] and Flowers[1] to further validate out of domain generalization and Kodak [3] true images for demonstrating high-resolution image fitting.

## 4.2 Training STRAINER'S shared encoder

We first trained separate shared encoder layers of STRAINER on 10 train images from each dataset. We share five layers, and train a separate decoder for each training image. For each dataset, we trained the shared encoder for 5000 iterations until the model acheives PSNR $\approx 30dB$ for all training images. We use the resulting encoder parameters as initialization for test signals in the following experiments. For comparison, we also trained the Meta-learned 5K baseline using the implementation provided by Tancik et.al.[45] with 5000 outer loop iterations. We also use the implementation provided by IPC[23] as our baselines for TransINR[9] and IPC[23] and train them with 14,000 images from CelebA-HQ . We report a comparison of number of training images and parameters, gradient updates, and learning time in Table 5.

## 4.3 Image fitting (in-domain)

We first evaluated STRAINER on the task of in-domain image fitting. We cropped and resized all images to $178 \times 178$ and ran test-time optimization on all models for 2000 steps.

At test-time, both STRAINER and STRAINER-10 use only 1 decoder, resulting in the same number of parameters as a SIREN INR. Table 1 shows average image metrics for in-domain image fitting reported with 1 std. deviation. Instead of naively fine tuning using another INR, STRAINER's design of sharing initial layers allows for learning highly effective features which transfer well across images in the same domain, resulting in high quality reconstruction across CelebA-HQ and AFHQ and comparable to Meta-learned 5K for OASIS-MRI images. Table 3(CelebA-HQ , ID) also shows that STRAINER initialization results in better quality reconstruction, when optimized at test-time, compared to more recent transformer-based INR approaches such as TransINR and IPC as well.

Table 1: **In-domain image fitting evaluation.** STRAINER's learned features yield powerful initialization at test-time resulting in high quality in-domain image fitting

| Method | CelebA-HQ | | AFHQ | | OASIS-MRI | |
|---|---|---|---|---|---|---|
| | PSNR↑ | SSIM↑ | PSNR↑ | SSIM↑ | PSNR↑ | SSIM↑ |
| SIREN | 44.91 ± 2.13 | 0.991 ± 0.007 | 45.11 ± 3.13 | 0.991 ± 0.005 | 53.03 ± 1.72 | 0.999 ± 0.0002 |
| SIREN fine-tuned | 51.11 ± 3.16 | 0.997 ± 0.013 | 53.07 ± 3.47 | 0.997 ± 0.001 | 58.86 ± 4.12 | 0.999 ± 0.0012 |
| Meta-learned 5K | 53.08 ± 3.36 | 0.994 ± 0.053 | 53.27 ± 2.52 | 0.996 ± 0.044 | 67.02 ± 2.27 | 0.999 ± 0.0000 |
| STRAINER (1 decoder) | 50.34 ± 2.81 | 0.997 ± 0.001 | 51.27 ± 2.94 | 0.997 ± 0.001 | 57.76 ± 2.19 | 0.999 ± 0.0001 |
| STRAINER-10 | 57.80 ± 3.46 | 0.999 ± 0.001 | 58.06 ± 3.75 | 0.999 ± 0.001 | 62.80 ± 3.17 | 0.999 ± 0.0003 |

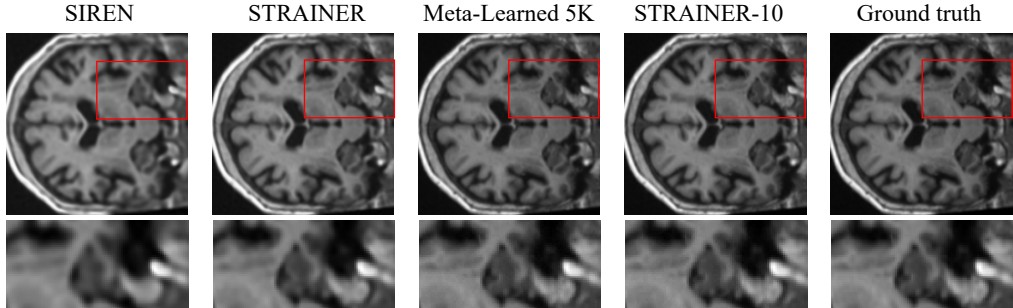

Figure 5: **Fitting MRI images from OASIS-MRI dataset.** At just 100 iterations, STRAINER is able to represent medical images with high quality. STRAINER's initialization allows for fast recovery for sparse and delicate structures, showing applicability in low-resource medical domains as well.

Table 2: **Out of domain image fitting evaluation, when trained on CelebA-HQ and tested on AFHQ and OASIS-MRI.** STRAINER's learned features are a surprisingly good prior for fitting images out of its training domain.

| Method | AFHQ | | OASIS-MRI | |
|---|---|---|---|---|
| | PSNR↑ | SSIM↑ | PSNR↑ | SSIM↑ |
| Meta-learned 5K | 52.40 ± 4.21 | 0.991 ± 0.077 | 65.06 ± 1.04 | 0.999 ± 0.00001 |
| STRAINER-10 | 57.46 ± 3.39 | 0.999 ± 0.0003 | 72.21 ± 8.73 | 0.999 ± 0.0001 |
| STRAINER-10 (Gray) | – | – | 74.61 ± 9.96 | 0.999 ± 0.0003 |

Table 3: **Baseline evaluation for image-fitting for in-domain(ID) and out-of-domain(OD) data.** STRAINER learns more transferable features resulting in better performance across the board. Models trained on CelebA-HQ unless mentioned otherwise. TTO = Test time optimization.

| Method | CelebA-HQ (ID) PSNR↑ | AFHQ (OOD) PSNR↑ | OASIS MRI (OOD) PSNR↑ |
|---|---|---|---|
| Meta-learned 5K | 53.08 | 52.40 | 55.86 |
| Trans INR w/o TTO | 31.59 | 28.63 | 31.97 |
| Trans INR w TTO | 51.86 | 49.01 | 55.45 |
| IPC(ReLU + Pos Enc.) w/o TTO | 33.27 | 29.96 | 33.96 |
| IPC(ReLU + Pos Enc.) w TTO | 49.72 | 47.19 | 51.35 |
| STRAINER-10 | **57.80** | **57.46** | 59.50 |
| STRAINER-10 ( trained on Flowers[1]) | - | 56.98 | 58.52 |
| STRAINER-10 ( trained on StanfordCars[2]) | - | 56.88 | **59.66** |

## 4.4 Image fitting (out-of-domain)

To test out-of-domain transferability of learned STRAINER features, we used STRAINER-10 's encoder trained on CelebA-HQ as initialization for fitting images from AFHQ (cats) and OASIS-MRI datasets (see Table 2). Since OASIS-MRI are single channel images, we trained Meta-learned 5K and STRAINER-10 (GRAY) on the green channel only of CelebA-HQ images. To our surprise, we see STRAINER-10 and STRAINER-10 (GRAY) clearly outperform not only Meta-learned 5K , but also STRAINER-10 (in-domain). To further validate out of domain performance of STRAINER , we train

Table 4: **Out-of-domain image fitting on Kodak Images [3]**. SIREN (trained on CelebA-HQ ) allows better convergence comparable to high capacity SIREN models as indicated by PSNR metric.

| | | Parrot | | | Airplane | | | Statue | | |
|---|---|---|---|---|---|---|---|---|---|---|
| Method | Width | PSNR↑ | SSIM↑ | LPIPS↓ | PSNR↑ | SSIM↑ | LPIPS↓ | PSNR↑ | SSIM↑ | LPIPS↓ |
| SIREN | 256 | 36.77 | 0.94 | 0.13 | 31.89 | 0.87 | 0.19 | 34.68 | 0.94 | 0.093 |
| STRAINER-10 | 256 | 39.55 | 0.96 | 0.087 | 35.03 | 0.92 | 0.09 | 37.84 | 0.96 | 0.037 |
| SIREN | 512 | 40.18 | 0.96 | 0.11 | 34.23 | 0.90 | 0.14 | 38.80 | 0.97 | 0.051 |
| STRAINER-10 | 512 | 44.38 | 0.97 | 0.021 | 38.96 | 0.96 | 0.023 | 43.92 | 0.98 | 0.008 |

Table 5: **Training time and compute complexity.** We train all the methods for 5000 steps. STRAINER instantly learns a powerful initialization with minimal data and significantly fewer gradient updates.

| Method | # training images | # learnable params | Gradient updates / iteration | Time (Nvidia A100) |
|---|---|---|---|---|
| SIREN | N/A | 264,707 | N/A | N/A |
| STRAINER (1 decoder) | 1 | 264,707 | 264,707 | 11.84s |
| STRAINER-10 (10 decoders) | 10 | 271,646 | 271,646 | 24.54s |
| Meta-learned 5K | 10 | 264,707 | 794,121 ($\approx 3\times$more) | 1432.3s = 23.8 min |
| TransINR[9] | 14,000 | $\approx 40M$ | $\approx 40M$ | $\approx 1$ day |
| IPC[23] w TTO | 14,000 | $\approx 40M$ | $\approx 40M$ | $\approx 1$ day |

Table 6: STRAINER **accelerates recovery of latent images in inverse problems.** STRAINER captures an implicit prior over the training data allowing it to recover a clean latent image of comparable quality $3\times$ faster making it useful for inverse problems.

| | Super Resolution (Fast) | | Super-Resolution (HQ) | | Denoising | |
|---|---|---|---|---|---|---|
| Method | PSNR | # iterations | PSNR | # iterations | PSNR | # iterations |
| SIREN | 32.10 | 3329 | 32.10 | 3329 | $26.75 \pm 1.67$ | $203 \pm 66$ |
| STRAINER -10 | 31.56 | 1102 ($\approx 3 \times faster$) | 32.43 | 3045 | $26.41 \pm 1.39$ | $76 \pm 27$ |

STRAINER-10 's shared encoder on simply 10 images from Flowers[1] and Stanford Cars[2] datasets which have different spatial distribution of color and high frequencies than AFHQ and OASIS-MRI. For fair comparison, all models in Table 3(OOD) were fit with 3-channel RGB or concatenated gray images in case of OASIS-MRI. As shown in Table 3(OOD), STRAINER-10 provides superior out of domain performance for AFHQ trained on CelebA-HQ , followed by Flowers and Stanford Cars. For OASIS-MRI, we see STRAINER-10 having best performance when trained with StanfordCars. This result suggests that STRAINER is capable of capturing transferable features that generalize well across natural images.

STRAINER also fits high resolution Kodak[3] images well and is comparable to SIREN networks with twice the network width.

## 4.5 Inverse problems: super-resolution and denoising

STRAINER provides a simple way to encode data-driven priors, which can accelerate convergence on inverse problems such as super-resolution and denoising. We sampled 100 images from CelebA-HQ at $178 \times 178$ and added $2dB$ of Poisson random noise. We report mean values of PSNR achieved by STRAINER and SIREN models along with the iterations required to achieve the values. For super-resolution, we demonstrate results on one image from DIV2K[4, 47], downscaled to $256 \times 256$ for a low resolution input. We used the formulation shown in Equation (1), with $\mathcal{A}$ set to a $4\times$ downsampling operation. To embed a prior relevant for clean images, we trained the shared encoder of STRAINER with high quality images of resolution same as the latent recovered image. At test time, we fit the STRAINER model to the corrupted image, following Equation (1) and recovered the latent image during the iteration. We report STRAINER's ability to recover latent images fast as well as with high quality in Section 4.5

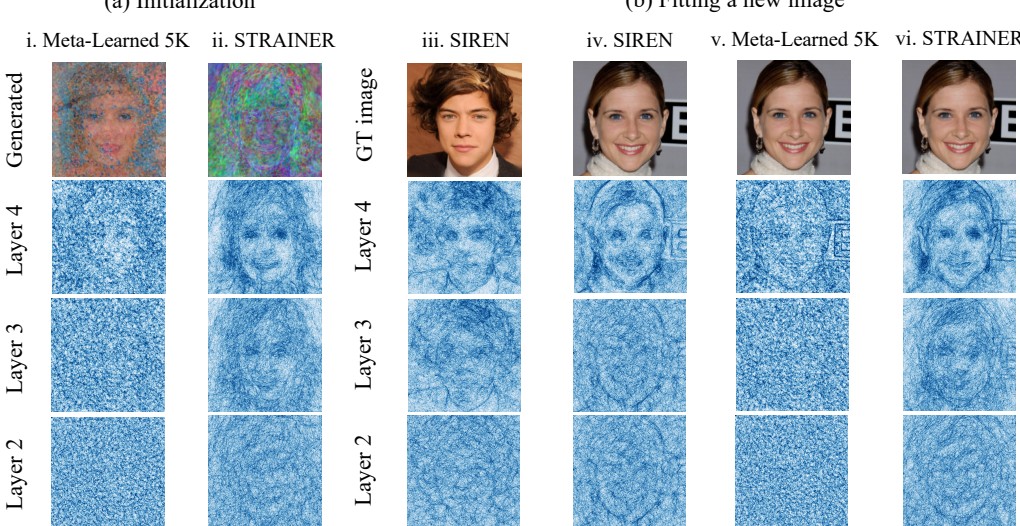

Figure 6: **Visualizing density of partitions in input space of learned models.** We use the method introduced in [20] to approximate the input space partition of the INR. We present the input space partitions for layers 2,3,4 across (a) Meta-learned 5K and STRAINER initialization and (b) at test time optimization. STRAINER learns an input space partitioning which is more attuned to the prior of the dataset, compared to meta learned which is comparatively more random. We also observe that SIREN (iii) learns an input space partitioning highly specific to the image leading to inefficient transferability for fitting a new image (iv) with significantly different underlying partitioned input space This explains the better in-domain performance of STRAINER compared to Meta-learned 5K , as the shallower layers after pre-training provide a better input space subdivision to the deeper layers to further subdivide.

## 5 Discussion and Conclusion

Results in Table 1, 3 demonstrate that STRAINER can learn a set of transferable features across an image distribution to precisely fit unseen signals at test time. STRAINER-10 clearly achieves the best reconstruction quality in terms of PSNR and SSIM on CelebA-HQ and AFHQ datasets, and is comparable with Meta-learned 5K on OASIS-MRI images. STRAINER-10 also fits images fast and achieves highest reconstruction quality than all baselines as shown in Figure 2. Comparing STRAINER (1 decoder) with a fine-tuned SIREN , it seems that the representation learned on one image is not sufficiently powerful. However, as little as 10 images result in a rich and transferable set of INR features allowing STRAINER-10 to achieve ≈7-10dB higher reconstruction quality than SIREN and SIREN fine-tuned.

As seen in Table 2, 3(OOD) STRAINER also performs well on out-of-domain tasks, which is quite surprising.

STRAINER's transferable representations are capable of recovering small and delicate structures as early as 100 iterations as shown in Figure 5 and do not let the scale of features from the training phase affect its reconstruction ability. Another interesting finding is that STRAINER-10 achieves far better generalization for OASIS-MRI (Table 2) when pretrained on CelebA-HQ . Further, STRAINER generalizes well to out-of-domain high-resolution images, as demonstrated by our experiments of training STRAINER on CelebA-HQ and testing on the Kodak data (see Table 4).

STRAINER is fast and cheap to run. Table 5 summarizes the time for learning the initialization for a 6 layered MLP INR for STRAINER , Meta-learned 5K and transformer-based methods such as TransINR and IPC. At 5000 iterations, STRAINER learns a transferable representation in just 24.54 seconds. Meta-learned 5K , in comparison, uses MAML[15] which is far more computationally intensive and results in $20\times$ slower runtime when exact number of gradient updates are matched. Further, STRAINER 's training setup is an elegant deviation from recent methods such as TransINR and IPC, requiring large datasets and complex training routines.

### 5.1 Limitations

Due to the encoder layers of STRAINER being tuned on data and the later layers being randomly initialized, we have observed occasional instability when fitting to a test signal in the form of PSNR "drops." However, we observe that STRAINER usually quickly recovers, and the speedup provided by STRAINER outweighs this issue. While our work demonstrates that INR parameters may be transferred across signals, it is not fully clear what features are being transferred, how they change for different image distributions, and how they compare to the transfer learning of CNNs and Transformers. Further work is needed to characterize these.

### 5.2 Further analysis of STRAINER

To further understand how STRAINER's initialized encoder enables fast learning of signals at test time, we explored the evolution of STRAINER's hidden features over iterations in Figure 3. In Figure 3(a), we visualize the first principal component of learned INR features of the STRAINER encoder and corresponding hidden layer for SIREN across iterations and observe that STRAINER captures high frequencies faster than SIREN. This is corroborated by the power spectrum inset plots of the reconstructed images. We also visualize a histogram of gradient updates in Figure 4, and observe that STRAINER receives large gradients in its encoder layers early on during training, suggesting that the encoder rapidly learns of low-frequency details.

Next, we visualize the input space partitions induced by STRAINER and the adaptability of STRAINER's initialization for fitting new signals. We use the local complexity(LC) measure proposed by Humayun et.al.[20] to approximate different pre-activation level sets of the INR neurons. For ReLU networks, the zero level sets correspond to the spatial location of the non-linearities of the network. For periodic activations, there can be multiple non-linearities affecting the input domain. In Figure 6 we present the zero level sets of the network, and in Supplementary we provide the $\pm\pi/2$ shifted level sets. Darker regions in the figure indicate high LC, i.e., higher local non-linearity. Figure 6 also presents partitions for the baseline models.

SIREN models tend to overfit, with partitions strongly adapting to image details. Since the sensitivity to higher frequencies is mapped to specific input partitions, when finetuning with SIREN , the network has to unlearn partitions of the pretrained image resulting in sub optimal reconstruction quality. When comparing Meta-learned 5K with STRAINER , we see that STRAINER learns an input space partitioning more attuned to the prior of the dataset, compared to Meta-learned 5K which is comparatively more random. While both partitions imply learning high-frequency details, STRAINER's partitions are better adapted to facial geometry, justifying its better in-domain performance.

## 6 Broader Impacts

STRAINER introduces how to learn transferable features for INRs resulting in faster convergence and higher reconstruction quality. We show with little data, we can learn powerful features as initialization for INRs to fit signals at test-time. Our method allows the use of INRs to become ubiquitous in data-hungry areas such as patient specific medical imaging, personalized speech and video recordings, as well as real-time domains such as video streaming and robotics. However, our method is for training INRs to represent signals in general, which can adopted regardless of underlying positive or negative intent.

## Acknowledgments and Disclosure of Funding

This work is partially supported by NIH DeepDOF R01DE032051-01, OneDegree CNS-1956297, IARPA WRIVA 140D0423C0076 and NSF grants CCF1911094, IIS-1838177, and IIS-1730574; ONR grants N00014- 18-12571, N00014-20-1-2534, and MURI N00014-20-1-2787; AFOSR grant FA9550-22-1-0060; and a Vannevar Bush Faculty Fellowship, ONR grant N00014-18-1-2047.

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

## Suplementary Material / Appendix

In our work, we increase the representation capacity of the INR by leveraging the similarity across natural images (of a given class). Since each layer of an INR MLP is an affine transformation followed by a non linearity, we interpret the INR as a function that progressively warps the input coordinate space to fit the given signal, in our case the signal being an image. Similar images when independently fit to their respective INRs capture similar low-frequency detail such as shape, geometry, etc. whereas high-frequency information such as edges and texture are unique to each INR. We propose that these low-frequency features from the initial layers of a learned INR are highly transferable and can be used as a basis and initialization while fitting an unseen signal. To that end, we introduce a novel method of learning our basis by sharing a set of initial layers across INRs fitting their respective images.

Our implementation can be found on [1]Google Colab.

## Understanding the effect of sharing encoder layers

We further investigate how the number of initial layers shared affects the quality of reconstructed image. We start by sharing $K = 1$ layer as the encoder, and $N - K$ layers in each decoder and vary $K$ from 1 to $N - 1$. $K = N$ is equivalent to simply fine tuning the INR based on all weights from a fellow model. We tabulate our results for image quality (PSNR) in a fixed runtime of 1000 iterations. We find that sharing all but the last layer results in the most effective capturing of our shared basis leading to higher reconstruction quality as seen in fig. 7. This also suggests that the last decoder of the INR is mainly responsible for very localized features. Further our work motivates further interest to sutdy the nature of the decoder layers itself.

We show the effect of sharing layers and resulting reconstruction quality. We use a 5 layered Siren model for this experiment. We fit a vanilla Siren model to an image and report its PSNR in fig. 7. Further, we train our shared encoder by sharing $K = 1$ layers and so on , until we share $K = N - 1$ layers.

We see that the reconstruction quality progressively increases by sharing layers.

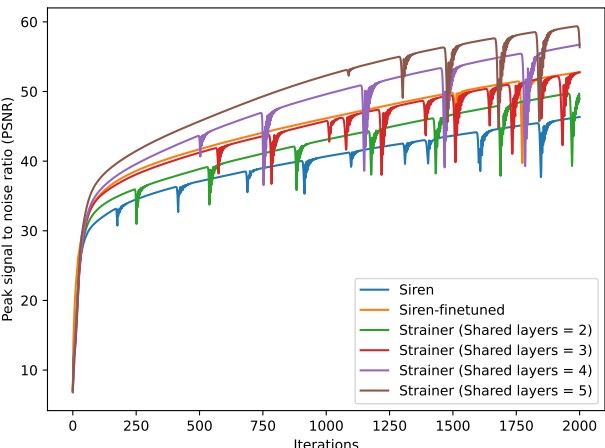

Figure 7: Sharing different number of layers in STRAINER's encoder. We see that by increasing the number of shared layers, STRAINER's ability to recover the signal also improves.

## Reporting std. deviation STRAINER for image fitting on CelebA-HQ

We also report the PSNR within 1 std. deviation while comparing STRAINER -10 with SIREN , SIREN -finetuned, STRAINER (1-decoder), and Meta-learned 5K in Figure 9.

---

[1]https://colab.research.google.com/drive/1fBZAwqE8C_lrRPAe-hQZJTWrMJuAKtG2?usp=sharing

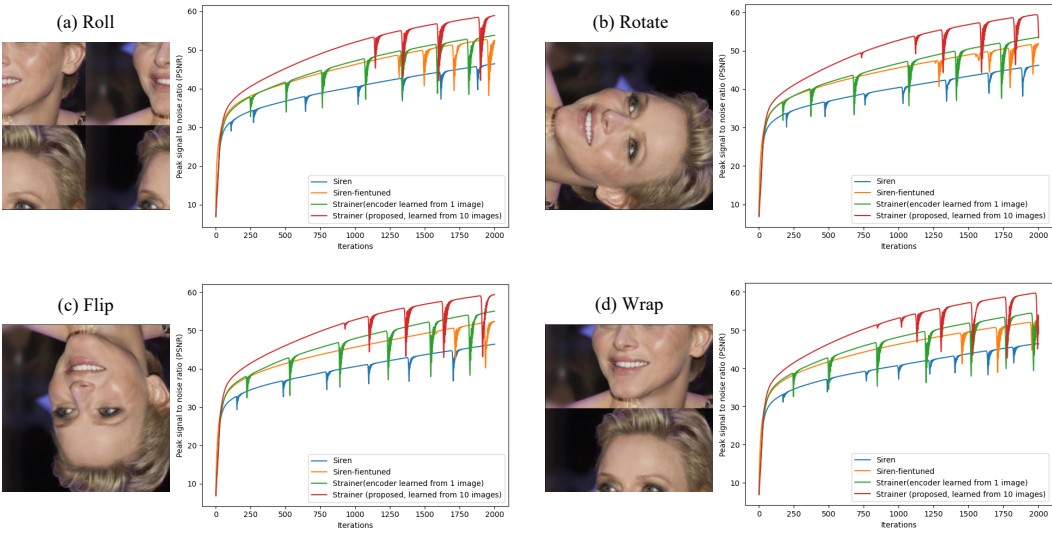

Figure 8: Fitting STRAINER on shifted, rotated and flipped versions of a face image. We see that despite the transformations done on a face image: (a) roll (wrapping the image both vertically and horizontally), (b) rotate, (c) flip vertically, and (d) wrap vertically, STRAINER fits equally well on all of them.

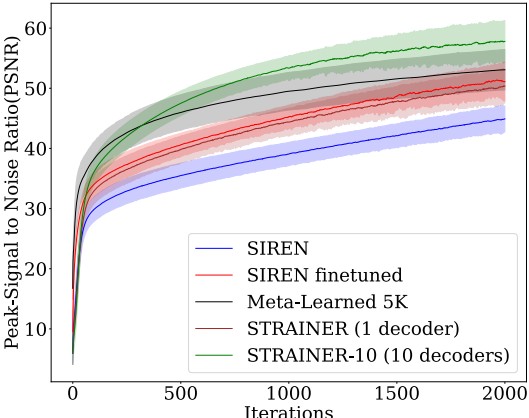

Figure 9: STRAINER learns fast. We show a limited baseline comparison of STRAINER with SIREN , SIREN -finetuned and Meta-learned 5K methods for the task of image-fitting on CelebA-HQ dataset and note that STRAINER achieves superior reconstruction quality.

## Effect of orientation of the input image

We wanted to further assess whether the INR is overfitting to one particular *aligned* face arrangement. To further test this, we take a test image and apply various augmentations such as flip, rotate, and *roll* and study how STRAINER fits it (see Figure 8). We find that the initialization learned by strainer is invariant, at test time, to the input signals orientation and can successfully capture the high frequency details fast.

## Measuring time for pretraining STRAINER and Meta-learned 5K

Our implementation is written in PyTorch[33] whereas Meta-learned 5K implemented by Tancik et.al[45] is a JAX implementation. For measuring runtime, we use the python's *time* package and very conservatively time the step where the forward pass and gradient updates occur in both methods. Further, we run the code on an Nvidia A100 GPU and report the time after averaging 3 such runs

for each method. There may be system level differences, however, to the best of our knowledge and observation, our timing estimates if not accurate are atleast indicative of the speedup provided by STRAINER .

## Training details for Kodak high resolution images

To further demonstrate that STRAINER's can be adapted to high resolution images, we evaluated our method on high quality Kodak[3] images with resolution $512 \times 768$ (see Tables 4 and 7). We present the reconstruction quality attained by STRAINER -10, SIREN model, and a SIREN model initialized using Meta-learned 5K , with widths of $256, 512$. For this experiment, we train our STRAINER encoder using CelebA-HQ Images which are resized to the same resolution to Kodak images. Further, we follow all steps as previously described for test-image evaluation of Kodak images. Here is another results from the Kodak high resolution images experiment.

Table 7: STRAINER allow better convergence comparable to high capacity Siren models, and meta-learned initializations, as indicated by PSNR metric. Tested on high quality Kodak Images. ID = In domain, OD= Out of domain.

| | Parrot (OD) | | | Airplane (OD) | | | Statue (OD) | | | Painted Face(ID) | | |
|---|---|---|---|---|---|---|---|---|---|---|---|---|
| Width=256 | PSNR↑ | SSIM↑ | LPIPS↓ | PSNR↑ | SSIM↑ | LPIPS↓ | PSNR↑ | SSIM↑ | LPIPS↓ | PSNR↑ | SSIM↑ | LPIPS↓ |
| SIREN | 36.77 | 0.94 | 0.13 | 31.89 | **0.87** | 0.19 | 34.68 | 0.94 | 0.093 | 32.03 | 0.85 | 0.26 |
| STRAINER-10 | **39.55** | **0.96** | 0.087 | **35.03** | 0.92 | **0.09** | **37.84** | **0.96** | 0.037 | **35.15** | **0.92** | 0.11 |
| Meta-learned 5K | 37.07 | 0.94 | **0.06** | 33.92 | 0.89 | 0.12 | 34.32 | 0.93 | 0.07 | 32.96 | 0.89 | 0.11 |
| Width=512 | PSNR↑ | SSIM↑ | LPIPS↓ | PSNR↑ | SSIM↑ | LPIPS↓ | PSNR↑ | SSIM↑ | LPIPS↓ | PSNR↑ | SSIM↑ | LPIPS↓ |
| SIREN | 40.18 | 0.96 | 0.11 | 34.23 | 0.90 | 0.14 | 38.80 | 0.97 | 0.051 | 34.45 | 0.90 | 0.17 |
| STRAINER-10 | **44.38** | **0.97** | 0.021 | 38.96 | 0.96 | 0.023 | **43.92** | **0.98** | **0.008** | **41.37** | **0.97** | **0.006** |
| Meta-learned 5K | 41.60 | 0.97 | **0.02** | **39.33** | 0.96 | **0.02** | 39.18 | 0.97 | 0.02 | 37.90 | 0.96 | 0.03 |

## Results for Inverse problems - Super Resolution

We discuss how STRAINER provides a useful prior for inverse problems such as super resolution. For the results reported in section 4.5, we attach supplementary plots as shown in fig. 10. STRAINER-10 (Fast) is a STRAINER-10 model with 5 shared encoder layers out of 6 total layers. STRAINER-10 (HQ) is a high quality STRAINER model with 3 shared encoder layers. Unlike forward fitting, more degree of randomness in the decoder helps recover better detail for inverse problems. We also showcase the effectiveness of STRAINER for in domain super resolution shown in fig. 11.

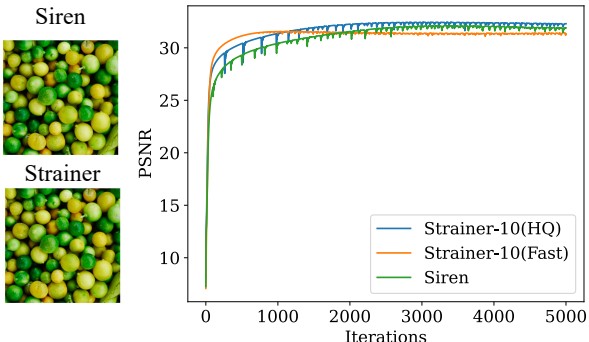

Figure 10: Super Resolution using STRAINER . We show the reconstructed results (a) on the left using SIREN and STRAINER . We also plot (b) the trajectory of PSNR with iterations. STRAINER-10 (Fast) achieves comparable PSNR to SIREN in approximately a third of the runtime.

## STRAINER for Occupancy fitting

STRAINER is a general purpose transfer learning framework which can be used to initialize INRs for regressing 3D data like occupancy maps, radiance fields or video. To demonstrate the effectiveness

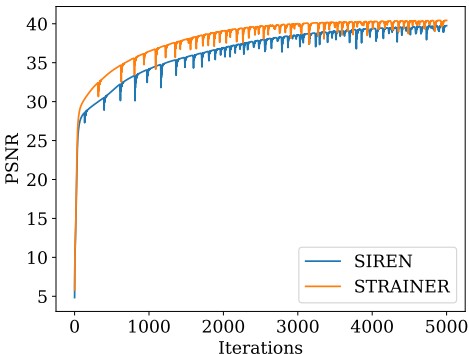

Figure 11: In domain $4\times$ super resolution using STRAINER . We see that STRAINER allows for faster convergence for in-domain super resolution making it useful especially for low time budgets. Max value achieved by STRAINER : $40.43dB$ while SIREN achieves $39.75dB$. Within 500 iterations STRAINER achieves $> 30dB$ PSNR

of STRAINER on 3D data, we have performed the following OOD generalization experiment. We pre-train STRAINER on 10 randomly selected 'Chair' objects from the ShapeNet[7] dataset. At test time, we fit the 'Thai Statue' 3D object[35]. STRAINER achieves a 12.3 relative improvement in IOU compared to random initialization for a SIREN architecture – in 150 iterations STRAINER-10 obtains an IOU of 0.91 compared to an IOU of 0.81 without STRAINER-10 initialization. We present visualizations of the reconstructed Thai Statue in Figure 12. Upon qualitative evaluation, we see that STRAINER-10 is able to capture ridges and edges better and faster than compared to SIREN.

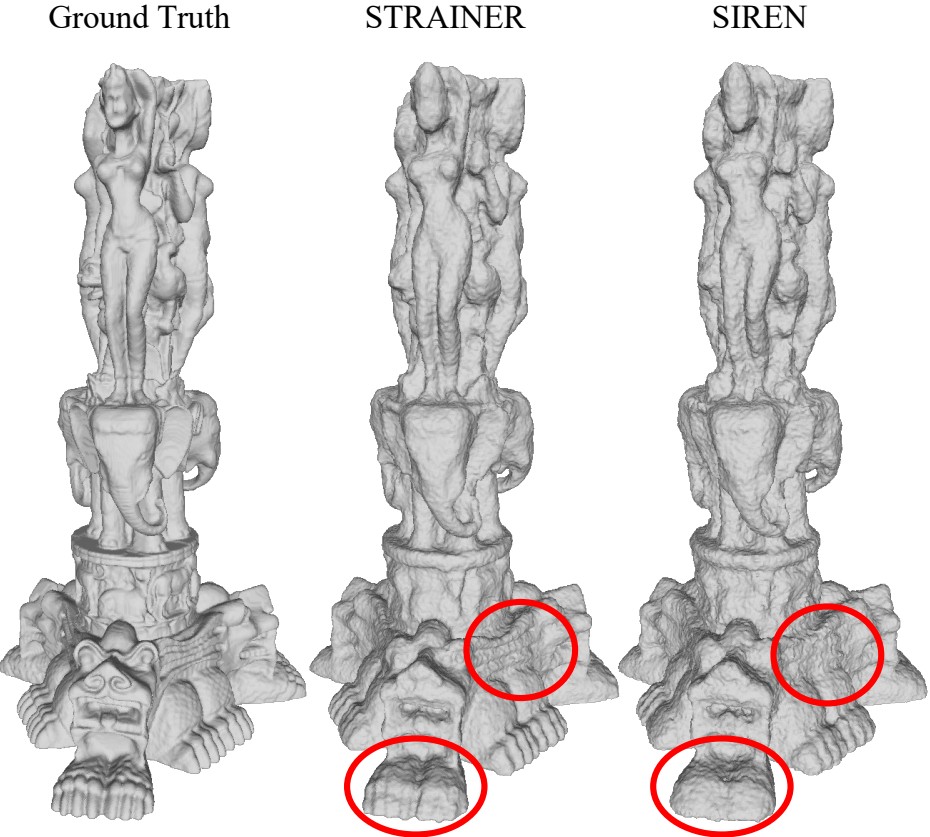

Figure 12: We use ten shapes from the chair category of ShapeNet[7] to train STRAINER , and use that initialization to fit a much more complex volume (the Thai statue[35]). We compare the intermediate outputs for both STRAINER and SIREN for 150 iterations to highlight STRAINER 's ability to learn ridges and high frequency information faster.

## Offsetting Pre-activations

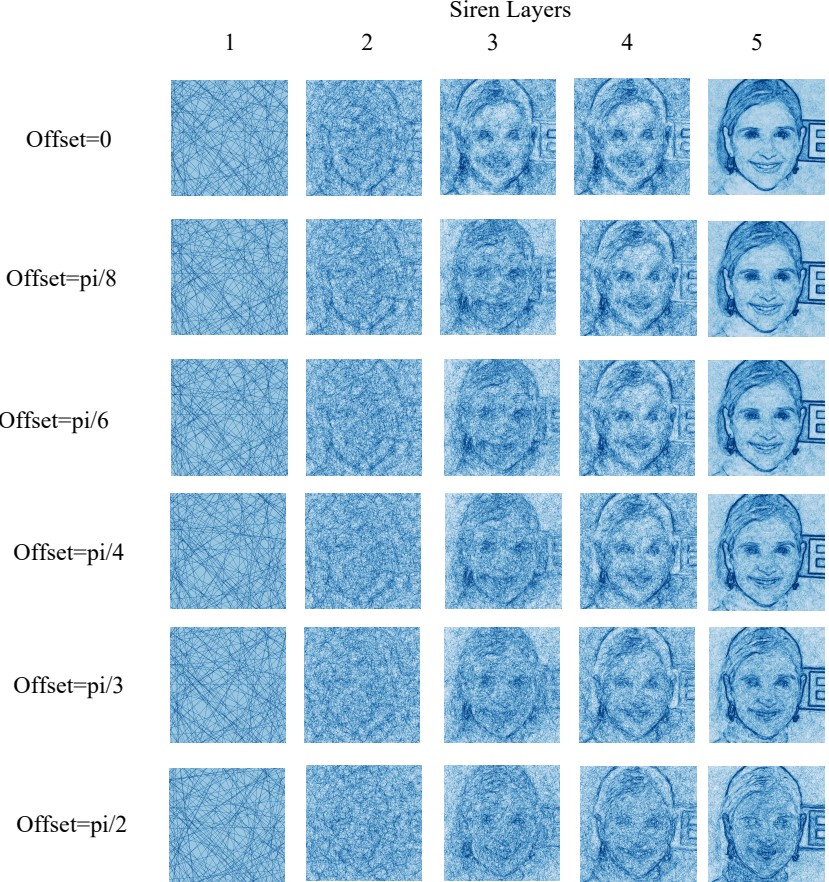

Figure 13: Siren offset

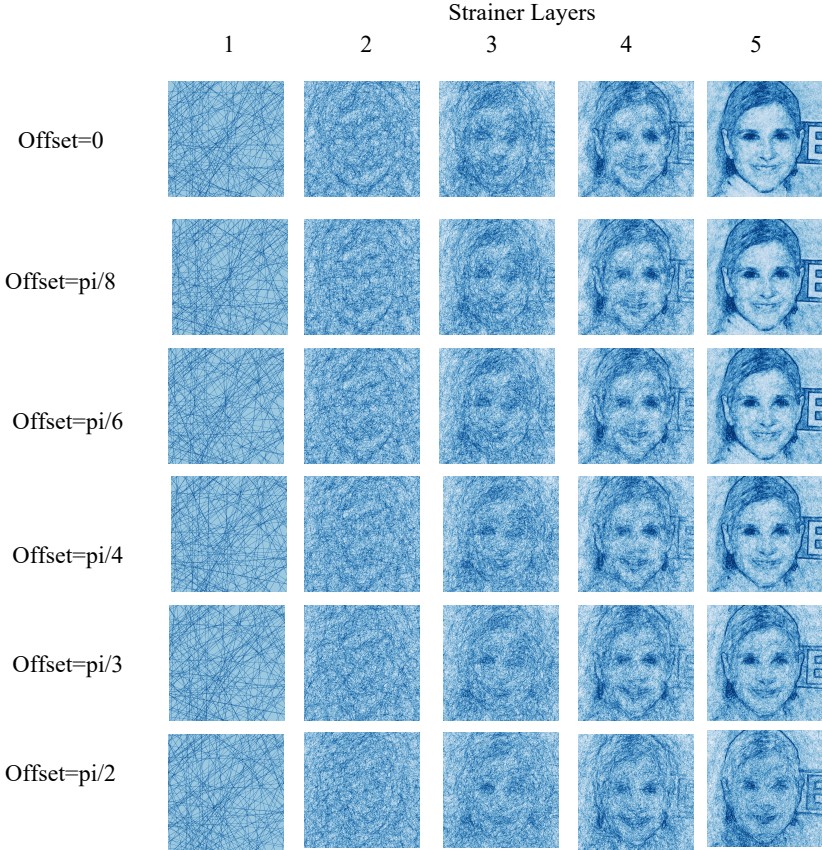

Figure 14: Strainer offset

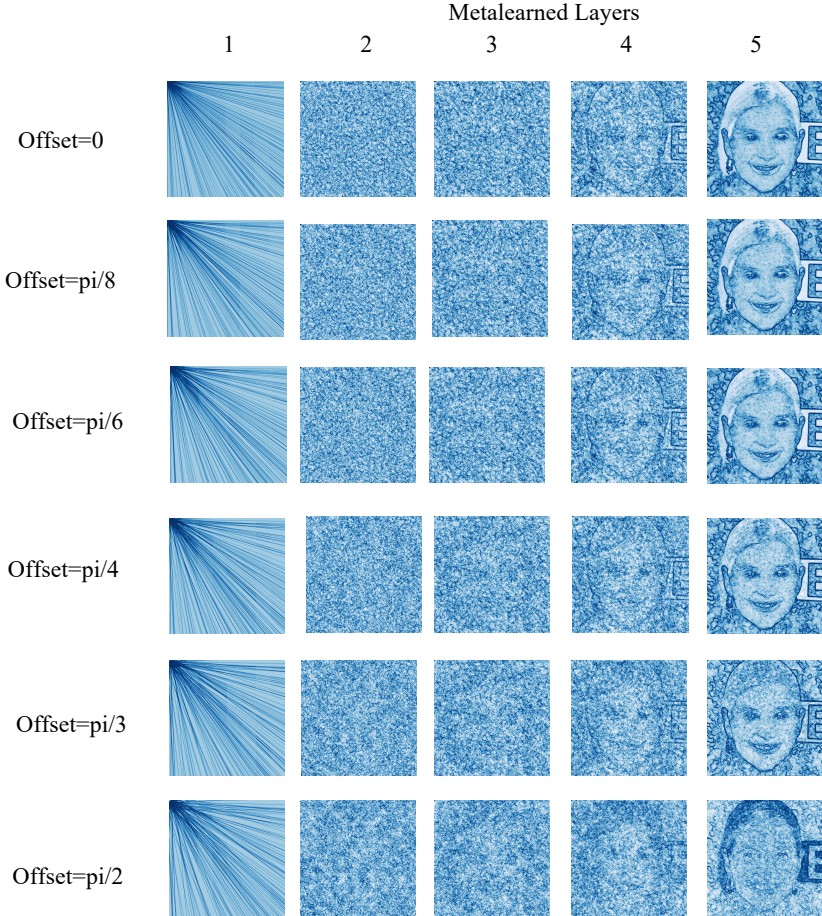

Figure 15: Meta-learned 5K offset

