# OpenReview forum: "Learning Transferable Features for Implicit Neural Representations"
_NeurIPS.cc/2024/Conference — NeurIPS 2024 poster_

### Official Review · Reviewer_GtRf · 2024-07-09

**Soundness:** 3
**Presentation:** 2
**Contribution:** 2
**Rating:** 6
**Confidence:** 4

**Summary:**

This paper introduces STRAINER, a novel training framework for Implicit Neural Representations (INRs). Unlike traditional INRs, which are trained on a single signal, STRAINER aims to learn transferable features that can be effectively utilized for fitting new signals from a similar distribution. The method involves sharing initial encoder layers across multiple INRs while using independent decoder layers. This setup enables STRAINER to achieve faster convergence and better reconstruction quality compared to baseline models. The paper evaluates STRAINER on various tasks, including image fitting, super-resolution, and denoising, demonstrating its effectiveness in both in-domain and out-of-domain scenarios. Detailed analyses are provided to understand the transferability of the learned features and their implications for INR training dynamics.

**Strengths:**

Originality:
The paper presents a novel approach to improving the transferability of INRs by sharing initial encoder layers. This is a significant departure from traditional methods that train INRs on a single signal without focusing on feature transferability.

Quality:
The empirical evaluations are thorough, covering multiple datasets and tasks such as image fitting, super-resolution, and denoising. The experiments convincingly demonstrate the advantages of STRAINER in terms of reconstruction quality and convergence speed.
Detailed analysis of training dynamics and feature transferability adds depth to the understanding of STRAINER's performance.

Clarity:
The paper is well-structured, with clear explanations of the methodology, experimental setup, and results. The inclusion of visualizations and tables enhances the readability and comprehension of the findings.

Significance:
STRAINER addresses a critical limitation in the current INR landscape by enabling feature transferability, which has broad implications for various applications, including medical imaging and inverse problems. This makes the work highly significant for the field.

**Weaknesses:**

Limited Theoretical Insights:
While the empirical results are strong, the paper could benefit from more theoretical insights into why the shared encoder layers facilitate better feature transferability. A deeper theoretical framework could strengthen the contributions.

Stability Issues:
The paper mentions occasional instability in the form of PSNR drops during test signal fitting. Although STRAINER recovers quickly, addressing this issue more comprehensively would improve the robustness of the method.

Comparison with Other Models:
The comparison with other advanced models, such as those using CNNs and Transformers for transfer learning, is not fully explored. Including more comparisons could provide a clearer picture of STRAINER's relative performance.

**Questions:**

1.Can the authors provide more theoretical insights or an intuitive explanation for why sharing initial encoder layers results in better feature transferability?
2.How does STRAINER perform when the distribution of training and test signals significantly differs? Are there specific characteristics of the signals that influence the transferability of features?
3.Could the authors elaborate on the stability issues observed during test signal fitting? Are there specific conditions or types of signals where these issues are more pronounced?
4.How does STRAINER compare with other transfer learning techniques that use CNNs or Transformers?

**Limitations:**

The authors have adequately addressed the limitations of their work, discussing the occasional instability observed during test signal fitting and the need for further characterization of the transferred features.

---

> ### Author Rebuttal · Authors · 2024-08-07
>
> We thank the reviewer for careful and thorough evaluation of our work. We appreciate the reviewer’s kind comments on STRAINER’s originality in learning transferable representations for INRs, extensive evaluation of STRAINER on datasets, and the detailed analysis provided by STRAINER on feature transferability and its training dynamics. We address the comments and suggestions below.
>
> __Theoretical insights: Why the shared encoder layers facilitate better learning.__
>
> Suppose we have an INR consisting of $L$ fully connected layers and non-linearity $\sigma$ after each layer. Assume that $\sigma$ is a piecewise-linear approximation of the sine function. For any piecewise linear non-linearity, the INR can be considered a continuous piecewise-affine operator, i.e., the INR subdivides the input space into pieces or regions $\omega \in \Omega$, where $\Omega_L$ is the input space partition due to all $L$ layers. Each region/piece of the function is mapped to the output in an affine manner.
>
> The subdivision $\Omega_L$, happens in a layer-wise fashion, e.g., layer $\ell$ contributes $\partial \Omega_\ell$ to the overall subdivision [13]. An important outcome of this layerwise subdivision is that shallower layers learn coarser partitions of the input domain, therefore resulting in shallower layers learning lower frequency features [55]. Deeper layers however can be more localized therefore, learn higher frequency or less symmetric features.
>
> The motivation behind INR is to learn a joint partition of the shallower layers, that can be further subdivided by the deeper layers into an optimal input space partition, resulting in a low error continuous piecewise-affine operator.
>
>
> __Unstable test time optimization for STRAINER.__
> We attribute the instability of STRAINER at test time fitting to the underlying SIREN model. In SP Figure 7 of the supplementary, reconstruction quality (PSNR) plots of Siren and STRAINER models plotted across time reflect the periodic instability. However, this can be fixed by using a learning rate (LR) scheduler.  As shown in RT Figure1(b), we apply an exponential multiplicative LR scheduler that decays the learning rate by 0.4 resulting in a smooth and stable learning for STRAINER-10 (shown in green).
>
> __Training distribution of images.__
> We conduct experiments in in-domain (ID) and out of domain (OD) image fitting where the shared encoder is trained on CelebA-HQ[16] and tested on CelebA-HQ, AFHQ[5], and OASIS-MRI[12,21]. We notice the superior performance of STRAINER compared to baselines (see RT Table 1) for in-domain image fitting. We also find that STRAINER captures highly transferable features. For OD image fitting on AFHQ, we find that STRAINER is able to reconstruct OD AFHQ almost comparable to ID fitting on AFHQ reconstructions, differing by a single digit in PSNR (MP Table 1,2). While for OASIS-MRI, we find STRAINER trained on CelebA-HQ surprisingly outperform even ID fitting on OASIS-MRI. To further assess the effect of training distribution on STRAINER, we train the shared encoder on Flowers [45] and Stanford Cars [46] as they have different spatial distribution of color and high frequencies compared to AFHQ and OASIS-MRI. We find that STRAINER is able to learn transferable features and achieves comparable PSNR to the shared encoder trained on CelebA-HQ[16]. As shown in RT Table 1, in-domain fitting for AFHQ[5] has the highest reconstruction PSNR, followed by STRAINER trained on CelebA-HQ, Flowers, and Stanford Cars (in decreasing order, presented in RT Table 1). The results suggest that STRAINER trained on natural images yields highly transferable representations, that inturn generalize well across natural images.  We propose this as a direction for future work.
>
> __Comparison with other transfer learning techniques.__ We compare STRAINER with other metalearning and transformer baseline models such as Meta-Learned 5K[37], TransINR[47] and IPC[48] (with and without test time optimization)  for image fitting. We evaluate STRAINER and baselines on CelebA-HQ (in-domain) , AFHQ(out of domain) and OASIS-MRI(out of domain). As seen in RT Table 1, STRAINER significantly outperforms IPC[48], TransINR[47] and other baselines by  ~ 3-5+ db in PSNR. We see that pretrained TransINR and IPC yield good PSNR at iteration 0, STRAINER catches up within little time and then consistently outperforms up until the end of 2000 iterations.
>
> We refer the reviewer to a more detailed summary of baseline results in our response to Reviewer FYQ8, “Comparison of baselines”, and “Metalearning baseline for Kodak”.
>
> Additionally, please also refer to our response to Reviewer HDgr, “New Out-of-Distribution generalization experiments”  for detailed evaluation of out of domain generalizability.
>
> Due to limited time, our experiments have been carefully chosen to address the concerns of all reviewers and are indicative of STRAINER learning highly transferable features for tasks such as image representation, with minimum training time and as little as 10 images. A comparison of training statistics and model footprints is mentioned in RT Table 3. We are happy to engage more and provide more findings during the discussion period.

---

> > ### Comment · Reviewer_GtRf · 2024-08-13
> >
> > I think the author has addressed the issues I raised.

---

> ### Author Response · Authors · 2024-08-13
>
> We sincerely thank the reviewer for their time, effort and their positive feedback in evaluating our work! We request the reviewer to kindly update their scores if our rebuttal experiments and analysis have satisfactorily addressed their comments and questions.

---

### Official Review · Reviewer_FYQ8 · 2024-07-10

**Soundness:** 1
**Presentation:** 3
**Contribution:** 1
**Rating:** 5
**Confidence:** 4

**Summary:**

This paper proposes STRAINER, where implicit neural representations (INRs) are trained for a class of similar objects, with the INRs for all objects having a shared, instance-agnostic encoder and instance-specific decoders. The paper also examines INR training dynamics. STRAINER is evaluated on 2D image regression and fitting MRI images.

**Strengths:**

Proposes the novel idea of sharing the first $K$ layers of the INR among all INR instances for a class of objects, while having instance-specific decoders for each instance, with some improvements in performance over SIREN/meta-learning baselines.

The authors include many experimental details so the experiments should be easy to reproduce.

The paper is well-written and easy to understand.

**Weaknesses:**

The paper only compares against one meta-learning baseline, ignoring the large amount of related literature (Trans-INR [1], Instance Pattern Composer (IPC) [2], PONP [3], Locality-aware generalizable implicit neural representation [4], etc.) that also looks at generalizable INRs. In particular, the results of this paper seem incongruent with the results of Instance Pattern Composers, which shows that modulation is most effective in the second layer of an INR (i.e., the second layer should be instance-specific while all other layers should be instance-agnostic).

The paper has limited experimental evaluation, limited only to image reconstruction and MRI reconstruction. The paper did not investigate the more complicated tasks that were also investigated by the baselines (e.g. LearnIt), such as novel view synthesis or CT reconstruction. Results on some tasks are also only comparable to the baseline (e.g. superresolution), and some tasks do not include all the baselines (e.g. the Kodak task does not have the LearnIt baseline).

No ablation study is done on some key hyperparameters, such as the number of shared encoder layers $N$.

In light of other works on generalizable INR, the contribution of this work is limited since it does not compare to many of the other works.

References
1. Chen, Yinbo, and Xiaolong Wang. "Transformers as meta-learners for implicit neural representations." European Conference on Computer Vision. Cham: Springer Nature Switzerland, 2022.
2. Kim, Chiheon, et al. "Generalizable implicit neural representations via instance pattern composers." Proceedings of the IEEE/CVF Conference on Computer Vision and Pattern Recognition. 2023.
3. Gu, Jeffrey, Kuan-Chieh Wang, and Serena Yeung. "Generalizable Neural Fields as Partially Observed Neural Processes." Proceedings of the IEEE/CVF International Conference on Computer Vision. 2023.
4. Lee, Doyup, et al. "Locality-aware generalizable implicit neural representation." Advances in Neural Information Processing Systems 36 (2024).

**Questions:**

1. While in a different setting, the IPC paper finds that the optimal instance-specific layer is the second layer of an INR. What accounts for the discrepancy between this paper's findings (the instance-specific layers should be the last few layers) and the findings of the IPC paper?
2. Why is the LearnIt baseline not used for the Kodak task?

**Limitations:**

Limitations are discussed in the paper. Negative societal impact does not seem to be a concern for this paper.

---

> ### Author Rebuttal · Authors · 2024-08-07
>
> We thank the reviewer for careful and thorough evaluation of our work. We appreciate the kind comments made on the novelty of our method and performance over Siren/meta-learning baselines. We welcome the reviewers suggestion on comparing with more baselines, and are thankful for the references provided and we will update them in Section 2, Background.
>
> __Comparison of baselines:__ Prompted by the primary concern of comparisons to baselines, we further conduct thorough image fitting comparison of STRAINER with IPC[48] and TransINR[47] with and without test time optimization as detailed below. As seen in RT Table 1, STRAINER significantly outperforms IPC[48], TransINR[47] and other baselines by  __~ 3-5+ db__ in PSNR. We attribute the superior performance of STRAINER to better input space partitioning learned by STRAINER as a consequence of sharing the initial layers, coupled with the instance specific decoder which provides an added degree of freedom allowing STRAINER to fit an unseen image with different morphology better. We see that pretrained TransINR and IPC yield good PSNR at iteration 0, STRAINER catches up within little time and then consistently outperforms up until the end of 2000 iterations.
>
> We also note that STRAINER features are more transferable for out of domain image fitting as shown for AFHQ and OASIS-MRI, even when the STRAINER encoder has been pre-trained on domains such as Flowers[45] and Stanford Cars[46]. In-domain fitting for AFHQ has the best reconstruction quality, followed by STRAINER trained on CelebA-HQ, Flowers, and Stanford Cars (in decreasing order, presented in RT Table 1.).
>
> __More details on our baselines.__ We use the publicly available implementation of IPC[48] which also includes the baselines for TransINR, both of which use FFNets[38]. While STRAINER uses Siren[31], we remark that FFNets and Siren models offer comparable representation capabilities for image fitting. We train TransINR[47] and IPC[48] baselines on 14,000 images from CelebA-HQ for 300 epochs[48]. To verify successful training of the model, we ensure that the PSNR on the test set is in near agreement with that reported in their respective papers barring changes due to the number of data samples.  We also ensure that all baselines and STRAINER MLPs have 6 layers and the same number of parameters. All models are evaluated on our test images for 2000 iterations. Lastly, for the OASIS-MRI dataset, we run all baselines using 3 channel MRI images and report metrics in RT Table 1.
>
> __STRAINER and IPC.__ In MP Figure 6.iv, we visualize the layerwise subdivisions / partitions in the input space of INRs. We observe that partitions evolve from coarse to fine as we go deeper in the INR. Initial layers provide for more global (low-frequency) transferable features. The deeper layers of the INR result in more complex subdivision of the input space giving rise to more local features and pertaining to high frequency detail in the image. As mentioned in MP Sec. 3.1, we motivate sharing layers in STRAINER to learn representations that give rise to partitions that generalize well across samples. IPC[48] proposes that the second layer in an INR be instance-specific while keeping all the remaining layers instance-agnostic. Modulating the second layer's weights would result in non-local changes resulting in suboptimal reconstruction. Further, IPC’s second layer weights obtained from a Transformer model may also affect the location of the optimal instance-specific layer.
>
> We further adapt IPC’s design of an instance specific second layer in STRAINER and compare it to other STRAINER models. As shown in RT Figure 1(b), we see the best reconstruction given by STRAINER-10(ours), then STRAINER (instance-specific second layer) and then SIREN - which validates our above intuition on input space division for STRAINER and IPC.
>
>
> __Metalearning baseline for Kodak:__ STRAINER learns higher quality reconstruction as indicated by image metrics (PSNR, SSIM[54], and LPIPS[53]) compared to Learnit baseline as shown in RT Table 2. We train all models on CelebA-HQ images of the same resolution as Kodak[1] images.  STRAINER performs better than SIREN and Meta-Learned 5K[37] with +2db in PSNR for network width of 256, and ~+3db for network width of 512. STRAINER(width=256) performs similar to SIREN(width=512), further attesting to the representation power of STRAINER’s learned features. We also remark that training time for STRAINER’s shared encoder on high resolution is negligible compared to meta learning models for high resolution image fitting - highlighting that STRAINER can be easily adapted to high resolution and out of domain images.
>
> __Ablation study on key hyperparameters:__
> We refer the reviewer to SP Figure 7. We show an ablation study on the number of layers shared in the STRAINER encoder and its ability to reconstruct the signal. We observe progressive increase in reconstruction quality (PSNR) as we increase the number of layers shared.
>
> __Evaluating STRAINER on complex (3D) signals.__ Please refer to our response to Reviewer HDgr, “Evaluating STRAINER on 3D signals”
>
> Due to limited time, our experiments have been carefully chosen to address the concerns of all reviewers and are indicative of STRAINER learning highly transferable features for tasks such as image representation, with minimum training time and as little as 10 images. A comparison of training statistics and model footprints is mentioned in RT Table 3. We are happy to engage more and provide more findings during the discussion period.

---

> > ### Comment · Reviewer_FYQ8 · 2024-08-12
> >
> > I appreciate the authors’ detailed response to the points raised by the reviewers. Most of my main concerns, such as comparisons to baselines such as IPC have been addressed by the authors, except the limitation that the method is mostly limited to signal fitting and is either inapplicable (e.g. NeRF) or has mixed performance (e.g. super-resolution, denoising) on inverse problems. Therefore, I would like to raise my rating to ‘borderline accept’.

---

> > > ### Author Response · Authors · 2024-08-13
> > >
> > > We sincerely thank the reviewer for their time and effort in evaluating our work. Indeed adapting STRAINER for 3D applications such as NERF is an interesting future direction for us. We greatly appreciate the positive assessment and feedback on our work!

---

### Official Review · Reviewer_HDgr · 2024-07-13

**Soundness:** 3
**Presentation:** 3
**Contribution:** 3
**Rating:** 5
**Confidence:** 4

**Summary:**

The research focuses on learning transferable features using a SIREN model, where encoder and decoder sub-networks are utilized to optimize weights for encoding and decoding. The paper evaluates the main claims made in the abstract and introduction, confirming that the goals are clearly stated and the method effectively addresses them, supported by results. The study discusses fitting unseen signals using optimized encoder weights and highlights the fast convergence of STRAINER to low and high frequencies during training. Experiments involve using the SIREN MLP model with sinusoid nonlinearities, along with different versions of STRAINER compared to baselines, all with equal parameters for fair comparison. Datasets such as CelebA-HQ, AFHQ, and OASIS-MRI are used for experimentation, with the shared encoder of STRAINER trained on 10 images from each dataset for shared initialization in subsequent tests. The research also explores image fitting tasks in-domain and out-of-domain, demonstrating the effectiveness of STRAINER compared to other models, specifically showcasing superior performance in various scenarios like super-resolution and denoising, emphasizing the importance of learned transferable features.

**Strengths:**

- The paper introduces the concept of STRAINER, which demonstrates fast learning of signals at test time by capturing high frequencies efficiently.

- It highlights the adaptability of STRAINER's initialization for fitting new signals, showcasing better performance compared to Meta-learned 5K in learning input space partitioning.

- Improved in-domain performance with STRAINER compared to meta-learning methods for INRs, providing better input space subdivision that aids in faster convergence on inverse problems like super-resolution and denoising.

**Weaknesses:**

- The paper missed mentioning a couple of important papers on meta-learning INRs, say Functa [A] and Spatial Functa [B]. Especially the latter achieves good PSNR as well as classification accuracy, despite that the methods and the focus are different.

[A] E Dupont et al. Your data point is a function and you can treat it like one.
[B] M Bauer et al. Scaling Functa to ImageNet Classification and Generation.

- The method is only tested on image datasets but not 3D shapes or radiance fields, where INRs are more applied to.

**Questions:**

- Besides the improvement in PSNR, is the generalization ability expected to help with completion of the signal (i.e. inpainting)?

- It is mentioned that the encoder trained on CelebA-HQ works even better than in-domain pretraining on AFHQ (cats) and OASIS-MRI datasets. Have you tried out-of-distribution generalization trained on other datasets as source domain?

**Limitations:**

Limitations are discussed in Section 5.1.

---

> ### Author Rebuttal · Authors · 2024-08-07
>
> We thank the reviewer for the careful and thorough evaluation of our work. We appreciate the kind comments on STRAINER being able to learn fast at test time, efficient recovery of high frequency components, and its ability to learn more transferable input space subdivision. We address the comments and suggestions below.
>
>
> __Citing additional papers on meta-learning INR.__ We thank the reviewer for the additional references. We find them relevant to our literature review and will include them in *Section 2, Background*.
>
>
> __Evaluating STRAINER on 3D signals.__ STRAINER is a general purpose transfer learning framework which can be used to initialize INRs for regressing 3D data like occupancy maps, radiance fields or video. To demonstrate the effectiveness of STRAINER on 3D data, we have performed the following OOD generalization experiment. We pre-train STRAINER on 10 randomly selected ‘Chair’ objects from the ShapeNet[44] dataset. At test time, we fit the ‘Thai Statue’ 3D object[52]. STRAINER achieves a 12.3% relative improvement in IOU compared to random initialization for a SIREN architecture – in 150 iterations STRAINER-10 obtains an IOU of 0.91 compared to an IOU of 0.81 without STRAINER-10 initialization. We present visualizations of the reconstructed Thai Statue in RT Figure 1(c). Upon qualitative evaluation, we see that STRAINER-10 is able to capture ridges and edges better and faster than compared to SIREN. We will add the results in the main text in Sec 4. and compare with baselines LearnIt[37], and IPC[48].
>
>
> Fitting signals (2D, 3D shapes, video) to STRAINER is straightforward and intuitive. However, extending STRAINER to inverse problems in 3D such as learning Neural Radiance Fields from limited observations and 3D Gaussian Splatting is non-trivial. We consider this an exciting future direction for us and thank the reviewer for their kind suggestions.
>
>
> __New Out-of-Distribution generalization experiments.__ Following the recommendations by the reviewer, we have trained STRAINER’s shared encoder on Flowers [45] and Stanford Cars [46] datasets and report reconstruction quality (PSNR) when testing on AFHQ [5] dataset in RT Table 1. We specifically chose Flowers [45] and Stanford Cars [46] as they have different spatial distribution of color and high frequencies compared to AFHQ and OASIS-MRI which would allow us to evaluate out of domain image fitting more thoroughly. We find that STRAINER is able to learn transferable features and achieves comparable PSNR to the shared encoder trained on CelebA-HQ[16]. As shown in RT Table 1, in-domain fitting for AFHQ has the highest reconstruction PSNR, followed by STRAINER trained on CelebA-HQ, Flowers, and Stanford Cars (in decreasing order, presented in RT Table 1). The results suggest that STRAINER is capable of capturing transferable representations that generalize well across natural images. We propose this as a direction for future work.
>
>
> We also emphasize in RT Table 2 (and MP Table 3 ) where we train on CelebA-HQ but evaluate out of domain on high resolution Kodak Images[1] on categories of airplane and statue where STRAINER outperforms baseline models substantially or is very comparable. Further, we  also compare STRAINER with recent work on generalizable INRs [47, 48].  We show in RT Table 1,  that STRAINER outperforms meta-learning as well as transformer baselines such as TransINR[47] and IPC[48], for out of domain fitting for AFHQ and OASIS-MRI datasets even when STRAINER is trained on unrelated source domains such as stanford cars.
>
>
> Due to limited time, our experiments have been carefully chosen to address the concerns of all reviewers and are indicative of STRAINER learning highly transferable features for tasks such as image representation, with minimum training time and as little as 10 images. A comparison of training statistics and model footprints is mentioned in RT Table 3. We are happy to engage more and provide more findings during the discussion period.
>
>
> __Generalization ability of STRAINER for Inpainting.__ We thank the reviewer for raising this important point. Successful recovery for inverse problems such as single-image inpainting relies on the inductive bias of the model. STRAINER’s rich representation encoded in the shared initial layers of an INR captures a prior over the data allowing it to converge rapidly fast. However, since it’s based on SIREN, its inductive bias is the same as siren - which may result in similar performance as Siren. We expect results similar to denoising and super-resolution as reported in MP Table 5 depending on the fidelity of the available signal. Further, for a non-trivial problem such as inpainting, conventional INRs may require support from added regularization terms if lacking strong inductive biases that favor solving inpainting, as shown by INRR [49].  Additionally, convolutional architectures such as the Deep Image Prior [50] leverage self-similarity and are successful at inpainting. While INRs do not exhibit locality bias, models such as Wire[51] have strong inductive bias due to the Gabor Wavelet which favors solving inverse problems.  We’re curious to assess if STRAINER can capture any favorable properties which can benefit the task of inpainting and will explore it as future efforts to understand STRAINER deeply.

---

> > ### Comment · Reviewer_HDgr · 2024-08-12
> >
> > Thanks for the response.
> >
> > I wonder if Spatial Functa should be added as a baseline in the experiments in rebuttal PDF Table 1. In terms of PSNR this method is also competitive.

---

> > > ### Author Response · Authors · 2024-08-12
> > >
> > > Functa[A] proposes to use neural fields (e.g. INRs) as data points itself for deep learning tasks such as classification and generation, but is limited in its ability to represent complex datasets such as ImageNet. Spatial Functa[B] addresses the limitation by using a spatially arranged latent representation of neural fields. At the core, both these papers use a separate _model_ to modulate the weights of an INR that represents an instance of the data, which is later used for downstream deep learning tasks. STRAINER (our work) provides a framework for learning powerful and transferable features for INRs, capturing a shared low-frequency across images by sharing initial _encoder_ layers of an INR and having signal-specific _decoder_ layers. Our work addresses feature transferability in INRs - enabling INRs to fit unseen in-domain and out-of-domain signals faster and with better quality. As you had previously mentioned and we agree, that Functa[A] and Spatial Functa[B] have different focus from our work (STRAINER). Further, since the code for Spatial Functa is not publicly available and due to limited available time in the discussion period, we find it unlikely to run additional baselines of Spatial Functa.
> > >
> > > Similar to Spatial-Functa, we find TransINR[47] and IPC[48] as reasonable comparisons since they too have separate _models_(transformer networks) that modulate the weights of the INR. We report our comparison in the rebuttal PDF RT Table 1. We show that while TransINR[47] and IPC[48] are trained using 14000 images for 1 day, STRAINER (our work) outperforms them significantly $\approx$ 5-7+db for in-domain and out of domain image fitting with just 10 training images and 24 seconds of training time. We also show a detailed comparison of training data size, training time and number of learned parameters in RT Table 3, and note that STRAINER uses orders of magnitude lesser training data and faster training time.
> > >
> > > We show extensive comparisons with other baselines in RT Table 1, and emphasize on STRAINER’s superior performance. We also show STRAINER on other modalities such as 3D occupancy maps.
> > >
> > > We request the reviewers to update their scores if our rebuttal experiments have addressed their questions and comments satisfactorily.

---

> > > > ### Comment · Reviewer_HDgr · 2024-08-12
> > > >
> > > > Thanks for the response. I still have some concerns about how the performance is compared with Spatial Functa, although the focus is slightly different. Spatial Functa should have good PSNRs, but I see that the code is not publicly available for a direct comparison. I will raise my score.

---

> > > > > ### Author Response · Authors · 2024-08-13
> > > > >
> > > > > We sincerely thank the reviewer for their time and effort in evaluating our work. We greatly appreciate the positive assessment of our work!

---

### Author Rebuttal · Authors · 2024-08-07

We are thankful to the reviewers for their careful and thorough evaluation of our work. We first provide key contributions of our work and then provide additional experiments and results as requested by reviewers.

__Summary of paper:__

STRAINER provides a framework to learn powerful and transferable features for implicit neural representations (INRs) by sharing a set of initial encoder layers of multiple INRs while being trained on a small set of prototypical signals (e.g. images, 3D occupancy). At test-time, for fitting an unseen signal, a STRAINER INR is initialized with the learned encoder and a randomly initialized signal-specific decoder. Our work addresses feature transferability in INRs - enabling INRs to fit unseen in-domain and out-of-domain signals faster and with better quality.

We empirically show that STRAINER learns a powerful representation from just 10 images and 24 seconds of training time (measured on Nvidia A100) and significantly outperforms models such as Siren and Meta-Learning based learned initializations[37] for image fitting - making it highly efficient for learning transferable features in data-scarce scenarios. Current data-driven learned initialization methods such as meta learning or hypernetworks rely on large models, big datasets, and long and potentially unstable training regimes[40].

Further, we provide detailed visualization of STRAINER’s learned representations, training dynamics and input space partitioning to further validate that STRAINER learns high frequency details faster and leads to better convergence. Using an approximation of SplineCAM[13], we visualize the input space partitioning which evidences STRAINER to learn more transferable features.

We also explore the nature of prior capture by STRAINER which leads to rapid or better convergence for inverse problems such as super resolution and denoising, and showcase its applicability for enabling INRs to learn powerful representations in medical imaging as well.


__Summary of Reviews.__
We thank the reviewers for their careful and thorough evaluation of our work, and for providing relevant references. Our reviewers have requested to conduct
- Detailed evaluation of baselines with recent INR literature such as TransINR[47], and IPC[48].
- Showcase STRAINER on various tasks beyond image fitting, such as 3D data and inpainting.
- Offer theoretical insights in how STRAINER learns transferable features and address the difference in STRAINER and IPC for generalization.
- address distribution of training data, out of domain generalizability of STRAINER, and stability of STRAINER while fitting a new signal


__Summary of additional experiments and analysis__

- We run multiple baselines with recent transformer based models such as TransINR[47] and IPC[48] for in-domain(ID) and out-of-domain(OD) image fitting and show that STRAINER outperforms across all baselines and tasks.
- We run OD experiments on 3 channel OASIS MRI images (compared to previously reported single channel MRI images) to facilitate fair comparison across newer baselines of TransINR and IPC.
- We show additional experiments where the STRAINER encoder is trained on different source datasets Flowers[45] and StanfordCars[46] and showcase the out of domain generalizability of STRAINER.
- We provide more detailed theoretical understanding on transferable features learned by STRAINER and address the difference in STRAINER and IPC for generalization.
- We expand STRAINER beyond just 2D images, inverse problems, and apply it on 3D occupancy maps showing STRAINER to learn 12.3% better in a limited number of iterations.

For conciseness and reading clarity, we define following acronyms to refer to figures and tables from their respective sources.

- RT : Refers to the Rebuttal PDF present in the author rebuttal.
- MP : Main paper
- SP : Supplementary material / Appendix attached to Main paper.


__References__

[44] A Chang et.al., ShapeNet: An Information-Rich 3D Model Repository, CORR 2015

[45] Kaggle, Flowers Dataset.

[46] Kaggle, Stanford Cars Dataset.

[47] Yinbo Chen et.al, Transformers as Meta-Learners for Implicit Neural Representations, ECCV 2022

[48] Chiheon Kim, Doyup Lee et.al, Generalizable Implicit Neural Representations via Instance Pattern Composers, CVPR 2023.

[49] Zhemin Li, et.al. Regularize implicit neural representation by itself, CVPR 2023.

[50] Dmitry Ulyanov et.al., Deep Image Prior, CVPR 2018

[51] Vishwanath Sargadam et.al, Wire : Wavelet Implicit Neural Representations, CVPR 2023

[52] Stanford 3D Scans Repository, Thai Statue.

[53] Richard Zhang et.al, The Unreasonable Effectiveness of Deep Features as a Perceptual Metric, CVPR 2018.

[54] Zhou Wang et.al., Image Quality Assessment: From Error Visibility to Structural Similarity, IEEE Trans. on Image Processing 2004.

[55] Randall Balestriero et.al., A Spline Theory of Deep Learning, PMLR 2018.

---

### Author Response · Authors · 2024-08-12

As we approach the end of the discussion period, we kindly urge the reviewers to update their scores if our rebuttal experiments have satisfactorily addressed their questions.

We sincerely appreciate the time and effort put in the careful evaluation of our work.

---

### Decision · Program_Chairs · 2024-09-25

**Decision:**

Accept (poster)

**Comment:**

All reviewer recommend acceptance as the paper proposes a novel idea for the transferability of INRs. After rebuttal reviewers were happy with the evaluations. Authors are encouraged to add the additional info in the final copy.